# Artificial Intelligence Techniques Used to Extract Relevant Information from Complex Social Networks

**DOI:** 10.3390/e25030507

**Published:** 2023-03-16

**Authors:** Santiago Paramés-Estévez, Alejandro Carballosa, David Garcia-Selfa, Alberto P. Munuzuri

**Affiliations:** 1Group of NonLinear Physics, University of Santiago de Compostela, 15706 Santiago de Compostela, Spain; 2Galician Center for Mathematical Research and Technology (CITMAga), 15782 Santiago de Compostela, Spain; 3CESGA (Supercomputing Center of Galicia), Avda. de Vigo s/n, 15705 Santiago de Compostela, Spain

**Keywords:** Twitter, complex networks, machine learning, CNN

## Abstract

Social networks constitute an almost endless source of social behavior information. In fact, sometimes the amount of information is so large that the task to extract meaningful information becomes impossible due to temporal constrictions. We developed an artificial-intelligence-based method that reduces the calculation time several orders of magnitude when conveniently trained. We exemplify the problem by extracting data freely available in a commonly used social network, Twitter, building up a complex network that describes the online activity patterns of society. These networks are composed of a huge number of nodes and an even larger number of connections, making extremely difficult to extract meaningful data that summarizes and/or describes behaviors. Each network is then rendered into an image and later analyzed using an AI method based on Convolutional Neural Networks to extract the structural information.

## 1. Introduction

The everyday use of social media leaves behind, as waste, vast deposits of data that can be exploitable to better understand social dynamics on the internet if used correctly [1,2]. In social media, people interact with each other building up communities through networks of friends and interactions, diffusing information, sharing ideas, and closing up physical barriers. The new age of Big Data offers new possibilities on how research is performed, posing new frameworks and key questions that are revolutionizing the field of computational social science [3]. Using tools from network science and mathematical models, we can transform all this data into rich insight on human behavior dynamics. In the past decade, many works have been devoted to analyzing data from social media sites such as Facebook, LinkedIn, or Twitter. Some examples include studies that have focused on understanding online personality patterns [4], usage behavior of social media [5,6] or its effect on human behavior and management [7,8,9,10]. In the field of rumor spreading, data from Twitter has been used to explore the opinion formation processes [11], opinion polarization [12,13,14], emergence of echo chambers [12,15,16,17], misinformation diffusion [18,19,20,21], or viral spreading of memes [22,23]. Other interesting works have measured sentiment metrics [24,25], state of opinions [26,27], or tracked the evolution of certain topics or conflicts [28,29,30].

Here, we implement an approach to transform daily information from *Twitter* into interaction networks between users. As an online plaza where constant debates are held and news ranging from the cultural spectra to political updates are commented on, *Twitter* is a rapidly changing network where trending topics have a mean lifetime of merely days [22]. New fresh topics appear quickly and the whole cascade of interactions resets and starts anew. We are interested in examining these interaction networks, how they change in time, and if they have relevant information from the structural point of view of network science [31]. In order to tackle this, we have computed properties that characterize their graph structure and that are directly related with the interaction patterns and information spreading, observing their time evolution from daily generated networks, with a time span of months. This task, however, had an important drawback since both the complexity of these calculations and the computational resources needed scale quickly with the network size. Thus, the task becomes impossible if we try to scale up our analysis. This can become quite unfeasible considering that we are talking about thousands and thousands of users posting on a daily basis, which results in heavy memory loads and considerably long computations. In order to overcome these handicaps, we additionally propose to introduce techniques based on artificial intelligence that can compute the same metrics in a quick and more efficient way. We used the parameters computed in the previous step and the visualization of the downloaded networks to train a convolutional neural network (CNN) to then make a prediction on those. We found that the CNN performed well with the test set, with most of the predictions falling within the confidence interval and the remarkable result that the computational times were reduced from minutes to milliseconds.

The manuscript is organized as follows. First, we describe the method to construct the social network of a single day and the protocol used to construct the hundreds of complex networks used along this manuscript. With this information, we define the structural parameters that characterized each network and present the convolutional neural network used to obtain this information using machine learning techniques. The results of our study are presented in the next section, and the manuscript ends with some conclusions.

## 2. Methods & Materials

### 2.1. Social Network Construction

*Twitter* is a microblogging website that allows users to post metadata composed of text, images, videos, sound, or a combination of the former, in the form of so-called *tweets*. In order to keep up with the content of other users, one user can follow another to read their daily *feed* that comprises what is being published by the people one is following. This establishes unidirectional links that begin to form a directed network of the followed (usually called “friends”) and followers. In addition, when a user posts a tweet, another user can interact with it adding a response (a *mention*) or forwarding it to the feed of their followers (with a *retweet* or a *quote*). These types of interactions are the ones we will be focusing on since they are the ones that can be retrieved back in time using the *Twitter API for Academic Research*, a free-access plan to the *Twitter* API destined for research-focused employees at academic institutions. The information on how the follower network has evolved in time would probably offer more insights on a user’s online behavior and the website’s structure, but, sadly, it is not available for download. You can only download the friends and followers of a given user at the time of the download.

For the data collection, we developed python code based on the library *tweepy,* which was crucial in order to translate raw data from the *Twitter* API to python data structures. Going back from June 2019 to March 2021, we searched the first five key words or hashtags that had been the most popular in Spain for each day, and then used our software to collect at least 10,000 interactions per topic (some examples of the Hashtags used are in the Appendix A). Then, we built a network for each day (being around 660 days the total analyzed) where users were placed as nodes and interactions as directed links by merging the datasets obtained from the hashtags. A concept of special importance for computing the network metrics is that of the network being *connected*. This means that there were no isolated nodes or disconnected small islands. For this, we made sure that some “bridging users” existed between the sub-networks given by the different hashtags that connected both sub-networks by having interactions with people collected from two different hashtags (details and some graphical examples are presented in the Appendix A). We did this by tracing the largest connected components of the whole network and filtering out the rest of the nodes. The reason behind merging different topics was an assumption so that the overall relevant activity of that day was captured in a statistical way along our network. After the network was built, all the content information of the tweets and their users was discarded so only the topology-related mathematical information was left, which was relevant for our study. Building a different network for each day (an example can be seen in Figure 1a), we collected a set of snapshots of the Spanish activity pattern on the website for almost a year and a half, then we measured our selected observables and observed how they evolved. Note that the contents of the hashtags used were not relevant, as we were mainly interested in gathering a significant number of users and observing how they interacted between themselves. In fact, the number of hashtags considered was enough to reach statistical significance, as the calculated parameters reached a plateau and became independent on additional users added to the network (via incorporating additional hashtags). This is observed in Figure 1d and discussed below.

### 2.2. Complex Network Analysis and Metrics of Interest

The structure of the interaction network was fully encoded in the *adjacency matrix*, where $A_{ij}=1$ if there existed an interaction from user i to user j, and 0 otherwise. We recovered the activity of each user by summing up their interactions in what was called the connectivity degree, or, simply, degree $k_{i}$ of each user. Since our network was directed, we distinguished between the out-degree $k_{i}^{out}=\overset{N}{\underset{j}{\sum }}A_{ij}$, the number of interactions produced by user i, and the in-degree $k_{i}^{in}=\overset{N}{\underset{i}{\sum }}A_{ij}$, which sums the number of interactions that user i received. This was a first-order measure of network centrality that distinguished how active the different nodes were. The mathematical form of the degree distribution $p\left(k\right)$ depended on the network, which, for real networks, usually followed an exponential law,
(1)$$p\left(k_{out}\right)=Ck_{out}^{-\gamma_{out}},\;\;\;\;\;\;p\left(k_{in}\right)=Ck_{in}^{-\gamma_{in}},$$
with γ, the scaling exponent usually ranged between 2 and 3. Networks that exhibit this type of distribution were termed *scale-free* [32] and were formed by a majority of nodes that had only a few connections, while a few nodes, named hubs, were involved in most of the interactions. In general, $\gamma_{out}$ may differ from $\gamma_{in}$, with the hubs of each distribution representing totally different concepts. The hubs of $k_{out}$ represented extremely active users that interacted heavily with others, while the ones of $k_{in}$ were popular or controversial users that were receiving a lot of attention. In any case, the difference in size between the smallest node to the largest hub could be of orders of magnitude, making notions such as the average degree make not much sense since there would be a very large deviation from the mean. The exponent γ would be then the quantity of interest as it described how the activity in the network was distributed. If the exponent was lower, closer to two, the degree distribution had a long tail, indicating that there would be more presence of hubs or high-activity users, while the distribution of interactions will be either narrower or with larger difference between the frequency of smaller nodes and the hubs if the exponent is higher. If one applied logarithms at both sides of Equation (1), the exponent γ was easily obtained as the slope of a linear fit (an example of these values and how they are calculated is plotted in Figure 1b,c). Note that the data marked with a lighter color were not used in the linear fits, as those points were endowed with larger uncertainty.

We were also interested in another important metric that involved the distance between nodes, meaning the minimal number of links needed to reach one user from another. This metric was called the *shortest path length* between a pair of nodes, and what we used in our study was the average shortest path length $\bar{\ell }$ of all the network. For this purpose, we used the functions of the python library *networkx* that found the shortest paths between every two nodes in the network through Dijkstra’s algorithm [33].

We considered an additional structural parameter that was related with the previous one. We calculated, for each of the networks considered, the so-called diameter of the network $\bar{\ell_{max}}$ that was defined as the largest of the shortest paths connecting every two nodes in the network. Figure 1e shows an example of such calculation. This parameter provided information about the centrality of the network that was some sort of measurement of the extent of the social dynamics of that particular day in our context (represented by the network considered).

### 2.3. Convolutional Neural Networks (CNN)

As we pointed out earlier, one of our main goals in this study is to show that we can make use of machine learning techniques to compute these network properties. For that purpose, all the information gathered from Twitter for each day is summarized in a single picture (such as that in Figure 1a). This is the input for a machine learning tool that should provide us with the structural parameters describing the network. The choice of CNN is mainly due to its particular suitability for obtaining the main properties or features of images by means of feature maps [34,35]. In addition, CNNs have already proven their capabilities in network classification problems [36].

Although an initial attempt was made to tackle the problem using directly as input the adjacency matrices of the networks, the process was too inefficient because of memory problems trying to handle the large sizes of the matrices, which could usually weight in the order of 10 gigabytes. Computing network properties also require the building up of the adjacency matrix, and it is important to note that large networks with lots of nodes also increase the computational time of the Dijkstra’s algorithm which, in its simplest form, scales as $O(\left|N^{2}\right|$). Instead, from the adjacency lists of the networks one can generate a graph visualization through the open-source software *Gephi* [37], which is already optimized and disposes of multiple parameters to display the networks clearly. These parameters are fixed for all the images, highlighting that we use a repulsive distribution of type ForceAtlas 2, a *scaling* of 0.3 and the *gravity* parameter is set to 2.0. In any case, preliminary tests were made with other types of networks such as DNN (Dense Neural Networks) and RNN (Recurrent Neural Networks), specifically LSTM and GRU but, as expected, acceptable results are not achieved with these neural types of networks.

We build our CNN with convolutional layers interspersed with pooling layers and activation layers, dedicated to discovering the patterns in the images. Then, another layer is used to transform two-dimensional information into one-dimensional (the *flatten* layer), and finally a last one is used to convert all the information into a list of n holes where each one takes a value between 0 and 1 (the *dense* layer, with 100 neurons). Since the values of the exponents of a scale free network range between 2 and 3, each number between these two is associated with a place in a list. To refer to that number, only its position is made to be worth 1, while the rest of the gaps are made zero. The size of this list will depend on the accuracy level desired. This accuracy, however, is not related to the uncertainty of the value, since it is not an adjustment by itself, but an estimation. In this work, the neural network was allowed to estimate exponents to the second decimal number. More than 100 of the calculated networks have an exponent outside the range 2<γ<3, as they cannot be considered to have a scale-free topology, including them would only impoverish the training of the network. Each of the 100 neurons in the final layer is associated with a label corresponding to an exponent between 2.01 and 3.00 (limits chosen to train the neural network with as few labels as possible). Using 41 neurons to expand the values that can be taken by the exponents predicted by the neural network to the whole interval (1.00, 5.00) gives predictions identical to those of the first case, so that both networks are equally valid. The only difference is the computational power needed to handle them.

A summary diagram of the neural network’s architecture is presented in the Appendix A.

The procedure followed in this work has been summarized in Figure 2. Once the data is collected from Twitter, the networks are constructed (and plotted using Gephi) and two paths are followed. First, we calculate the structural parameters characterizing the network using conventional/mathematical techniques. Additionally, secondly, we use the machine learning techniques described above to perform the same calculations in a much shorter time. The agreement is shown below. Our aim is, not only to reduce the computation time to calculate the network parameters, but also to prove the possibility of extracting them from pictures of networks, a non-trivial operation made possible by our proposed architectures for Deep Learning models.

## 3. Results

### 3.1. Social Network Analysis

We have gathered a total of 656 social networks from the time span between June 2019 and March 2021, with a daily average of N=17800±9300 nodes per network, which we assume is a reasonable enough size to conduct our analysis. Figure 1a shows, as an example, a visualization of one of the networks, where we have highlighted the different hashtags and see that they are distributed evenly among all the subgraphs (additional details showing the process to construct this graph are shown in the Appendix A). Panels (b) and (c) show the typical degree distribution that the obtained networks follow, which we fit in a logarithmic scale to a line in order to obtain the exponent γ as the slope. This figure (Figure 1b) corresponds with the in-degree calculations or the network, describing how users influence the others. The equivalent results, considering the out-degree distribution, are shown for the same network in Figure 1c. Note that the scaling exponents calculated for each case are significantly different ($\gamma_{out}=2.55$ and $\gamma_{in}=1.67$), as we anticipated earlier in Section 2.2. The in-connectivity presents a smaller density of very low connected individuals and a bigger presence of highly connected ones, resulting in a less sharp distribution and thus a smaller scaling coefficient. Differences between these values, therefore, show important differences in the connectivity patterns and overall activity of different days, which is the case in many of the networks we build up.

Figure 1d,e show how the scaling exponent of the network converges to a certain value as we expand the network increasing the number of hashtags considered. Note that the value has converged after five hashtags, which is the standard number of hashtags that we consider for constructing each network. We assume that the exponent γ gives us accurate and relevant information on the activity pattern of that day once this value has converged. Figure 1f illustrates the calculation of the other two parameters for this example network, the average shortest path length $\bar{\ell }$, and the diameter of the network $\bar{\ell_{max}}$.

As we mentioned in the Section 2, we built daily networks with the data collected from *Twitter* and computed the metrics explained in Section 2.2; the in-and-out scaling exponents, the average shortest path, and the network diameter for each one. In addition, in Appendix A, we also plotted the time evolution of the network size N in order to show that the size along the time series was comparable. At a first glance, we noticed significant differences between $\gamma_{in}$ and $\gamma_{out}$ and how they evolved in time (see Figure 3a,b). The $\gamma_{in}$ scaling exponents were closer to two, describing scale-free distributions with a longer tail and a larger difference in the inward-connectivity between the hubs (popular users in this case) and the smaller users. In time, we see that it is a very regular time series: noisy but with no important spikes. This could suggest that there is always a similar proportion between the number of accounts that receive lots of attention and the ones that do not, independently of the content. The $\gamma_{out}$ scaling exponent, however, was closer to three and showed increasing and decreasing local trends, with more abrupt spikes, indicating differences in how users engage with each other depending on the trends. This could make sense from the point of view of the hashtag’s content. There are relatively calm weeks where trending topics are usually related only to TV shows or to cultural and sport events, where, usually, there is an official account promoting the key words to which the users of its community engage with. There are others, however, rich in some hashtags related to moral, controversial, or political news that can persist longer since they appeal to users from many different communities. This would involve either an increase in the activity of smaller nodes (that were engaging more in the network) or an increase in the frequency of lower-activity nodes (that entered the hashtag but only left a couple of interactions), both of which result in the increase of the $\gamma_{out}$ exponent. Understanding exactly how the tweets’ content alter the network structure is, however, slightly beyond the scope of this work, since a more dedicated analysis with hands-on data would be more appropriate. Furthermore, we saw that it was very difficult to find in this series any trace of deterministic patterns, such as oscillations or chaos. For that, in case they do exist, it would be necessary to look at the evolution of the exponent over many years or even consider a different approach to the selection of data.

Attending to the evolution of the average shortest path (Figure 3c), it does not seem to follow any particular trend either. This ensures that the information flow of the graph stays very similar independently of the day we choose, showing that reaching one node from any other node of the network takes always less than five connections. In scale-free networks, hubs represent strong shortcuts that drastically reduce the distance between nodes, and it would make sense that the shortest path would not change much either, since the $\gamma_{in}$ scaling exponent showed that the size of popular users barely changed in time.

The yellow vertical line plotted in Figure 3 marks the date when mandatory confinement was ordered by the government in Spain due to the COVID-19 pandemic. This is a crucial historical landmark that one can assume had also an important impact on social networks as well. Observation of the figures shows some impact, more relevant in the $\gamma_{out}$ scaling exponent (the horizontal red lines plotted in Figure 3 show the average value of each parameter before and after 15 March 2020), but further analysis with filtered data and paid attention to the media content should be conducted to verify this point. Note that it is not casual that the scaling exponent $\gamma_{out}$ is more affected by this historical landmark. As we mentioned earlier, this significant increase of the exponent could indicate that users decided to engage more with the hashtags appearing right after the beginning of the pandemic, which related more to healthcare issues and overall concern about the situation.

All the data presented in Figure 3 is shown as calculated without any filtering. The solid lines correspond with the average values and the shadow areas mark the uncertainties. Figure 3a,b show the scaling exponents that are calculated via a linear fit of the connectivities of each node (uncertainty comes natural as the uncertainty in the fitted parameters). Figure 3c,d show the average shortest path length ℓ (thus, the uncertainty is just the dispersion of all the values) and the network diameter $\bar{\ell_{\max}}$ that is averaged over the ten tops (and the dispersion is also calculated among those).

### 3.2. Prediction of the CNN

In this section, we present the results as calculated using the CNN tool developed in the Section 2, following the scheme in Appendix A.

This work is focused on characterizing networks using only the information available in their visual representations. Two different layer activation functions have been used: ReLU for convolutional layers and Softmax for the dense layer. The cost function has been based on the cross-entropy (as it tries to categorize the network parameters) and the mean square error (as it also regresses the value associated with this parameter after categorization against the calculated value), and the Adam algorithm has been chosen as optimizer. Of the total 656 networks, some had to be discarded because of issues with their Image rendering, and only 480 have been used here with their corresponding parameters. From these 480, 70% have been used for training and 30% for testing the different trained models. From the trained dataset, 10% have been selected to validate the fit. Thus, a validation split of the trained data has been set as 0.1, and the model has been trained with 200 epochs. Data has not been filtered even when non-ideal images were spotted in the dataset.

CNN is chosen given its high performance in data acquisition from images and because acceptable results can be achieved from just a simple neural network architecture, using low resolution images and combining categorization and regression. With better image resolutions, filtering the data, and after training more complex neural network models, parameters could be obtained with even more precision.

In order to evaluate the robustness of the model, we evaluated the cost functions during successive training epochs. The plots are presented in Appendix A.

From each network, we generated an image of 256 × 256 pixels similar to Figure 1a, building a dataset with a total of 480 images. Some examples of the images used to train the CNN are shown in Appendix A. Figure 4 and Figure 5 show the performance of the neural network predicting the in and out scaling exponents as well as the average shortest path and the network diameter compared to the actual calculated parameters. At first glance, it appears that most of the values are in good agreement with the fitted exponents (within the margins of error), so the predictions seem fairly accurate, especially considering that the exponents used to train the network had uncertainties as large as those used to test the results. The large values for the uncertainties can be explained by the fact that the networks with which the exponents are fitted are fragments of much larger networks, meaning that the fit to a free scale is not perfect, and thus uncertainty is high.

Note that the model was able to acquire the ability to estimate the values of the parameters of complex networks through their images and obtain a value close to the one calculated analytically in a much faster way. Again, our objective in this work was to show that this type of neural network is a tool that has this capability, even with a very basic architecture, with hardly any optimization of its hyperparameters. Of course (but this is beyond the scope of this paper), with optimized and more conscientious training, values could be predicted much more accurately.

We compared the results between the original test values and their predictions in Figure 5. For each of the four parameters we presented (along the diagonal), the distribution of values was directly calculated from the test networks and using CNN. Note that the distributions were similar, meaning that the model was able to statistically reproduce the set of possible values in all four cases. In most events, the prediction and test were in good agreement (see antidiagonals in Figure 5), demonstrating the applicability of this procedure.

In general, a Spearman’s coefficient value of ρ=0.21 seemed to indicate a positive correlation. On closer inspection, the cases where the prediction failed were images in which all the nodes were agglomerated in a large mass, making it difficult to clearly discern the connections between them. Thus, the method seems to fail mainly when the image was not recognizable. Further improvement on the predictability could be obtained directly using more images of networks, i.e., with a broader time window of analysis or considering larger-size detailed images.

## 4. Discussion/Conclusions

We reconstructed a representative sample of the daily network of *Twitter* interactions in Spain for a time window of more than a year, with the purpose of reflecting the way society interacts and its temporal evolution. This required the development of software that automated the download of huge amounts of freely-available data from the *Twitter* API to construct the temporal evolution of the network. The tool designed to extract information from *Twitter* had been registered as a result of this work, and their registration numbers are *SC-35-2022* and *SC-36-2022*. The developed method to construct the networks consisted of adding the resulting subnetworks of the trending hashtags of the day. As the hashtags were chosen by popularity, this presented an example of a network that we considered representative of the total network. The fact that the exponent of the network converged to a given value as more subnetworks were added (Figure 1d) confirms this hypothesis.

Each of the networks obtained was characterized in terms of topological parameters, finding that the activity pattern persistently followed a scale-free distribution of interactions, as was already observed in previous analyses of this website [22,38]. This is, now, also confirmed for the Spain case over a significantly large period of time. Attending to the inwards direction of links ($\gamma_{in}$ Figure 3a), which reflected the popularity of users, we observed that the distribution did not change much in time, thus the presence of large popular hubs with similar sizes was roughly constant independently of the trend. On the contrary, the overall activity pattern reflected by the outward direction of the links ($\gamma_{out}$ Figure 3b), showed noticeable fluctuations around local trends in the time series. These fluctuations in the average activity did not, however, show an important effect on the average shortest path of the networks, since the latter mainly depended on the hubs. The other parameter that was tracked was the network diameter ($\bar{\ell_{\max}}$). The observation of Figure 3d showed some average increase in the values right after the considered temporal landmark (the beginning of mandatory confinement in Spain ordered by the government in Spain due to COVID-19 pandemic around 15 March 2020). Additionally, at the same time, the out-scaling exponent ($\gamma_{\mathrm{out}}$) exhibited a noticeable increase that remained along the rest of the analyzed days. The combination of both results signified the direction that the mandatory confinement could have taken to affect social networks and that our structural measurements seem to be able to detect it. It is important to note that the machine learning tool developed that reduces computational times by several orders of magnitude also successful identifies this phenomenon. Note that the increase of the scaling exponent $\gamma_{\mathrm{out}}$ was related to an increase in the connectivity of the average (less active) user at the expense of less singular individuals with very-high presences (the out-hubs) in the network. This was confirmed by the fact that the network diameter $\bar{\ell_{\max}}$ had also increased at the same time.

Thus, the technique presented here to build statistically relevant social networks proves to be relevant and able to describe and detect significant phenomena that affects social networks. We believe that our studies shown here demonstrate that it is possible to detect significant events in the studied society without the necessity to access the contents of the messages exchanged between the users, based on structural properties of the social networks. This is a novel idea that opens many future possibilities.

Additionally, machine learning tools were introduced to reduce the decompression time of the networks and the numerical computation of their exponents. Once the neural network was trained, it was possible to reduce these average times by a factor of 10,000 (from 1 min to 50 ms), although the greatest improvement was observed in the calculation of the other four properties. This work can be put in contrast with the work in [39], where they also focused on the calculation of the exponents that had been shown to be the most significant for the study of complex networks.

The potential of convolutional neural networks for their application in the analysis of complex networks from images of these networks was revealed. Although the creation of such images was not extremely practical due to current methodology, simply the ability to compress TBs of information into MBs makes the idea tremendously attractive because, although there was already the *parquet* format that achieves something very similar, to date, we have found no other way to directly use compressed files in this way. The technology developed along this manuscript opens the possibility of extensive studies using freely available data.

## Figures and Tables

**Figure 1 entropy-25-00507-f001:**
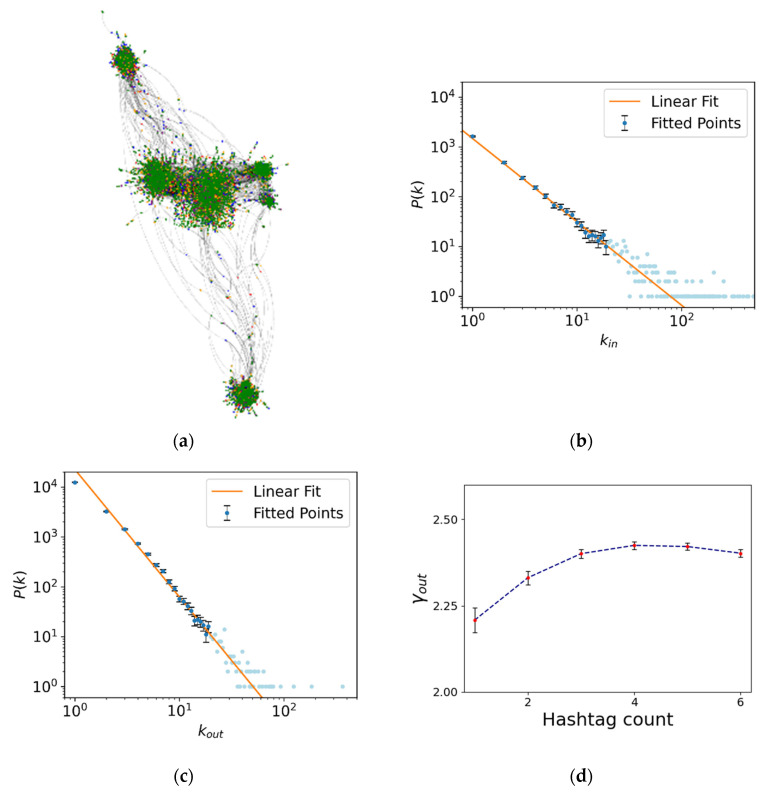
(**a**) Example of a social network visualized with *Gephi*, corresponding to 23 January 2020, where the colors represent the hashtag in which each user was more involved. (**b**) Example of a linear fit to the power law distribution of the in-degrees (with a calculated value of $\gamma_{out}=1.67$). Note that only the solid-colored points are the ones considered in the fit. (**c**) Example of a linear fit to the power law distribution of the out-degrees (with a calculated value of $\gamma_{in}=2.55$). Convergence of the scaling exponent (**d**) $\gamma_{out}$ and (**e**) $\gamma_{in}$ as we merge the different hashtags of the same day. The error bar represents the standard deviation of the fits among all the possible merging combinations. (**f**) Distribution of the mean distances for all the nodes in the same network as in panel (**a**). Note that $\bar{\ell }$ is just the mean value of these data (marked with the vertical green line) and $\bar{\ell_{max}}$ is the average of the 10 largest values (marked with the red vertical line).

**Figure 2 entropy-25-00507-f002:**
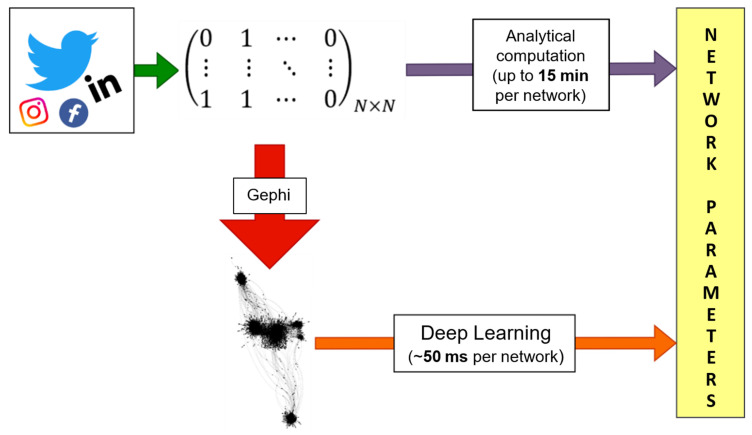
Flow chart of the process followed along the manuscript. Data can be extracted from any social media, in this work Twitter was chosen as source. From the data, in this work unfiltered to remove biases, adjacency matrixes can be built. The state-of-the-art method to calculate structural parameters consists of obtaining them analytically from the adjacency matrix. Our proposed method consists of using Gephi to turn said adjacency matrixes into images, much lighter than their matrix counterparts, which can be evaluated with the simple AI architecture we propose to obtain an approximation to the analytical values. This method, potentially, represents a way to compress data with dedicated computational units (generating images) to allow any device to calculate fast any parameter using the proper Deep Learning model.

**Figure 3 entropy-25-00507-f003:**
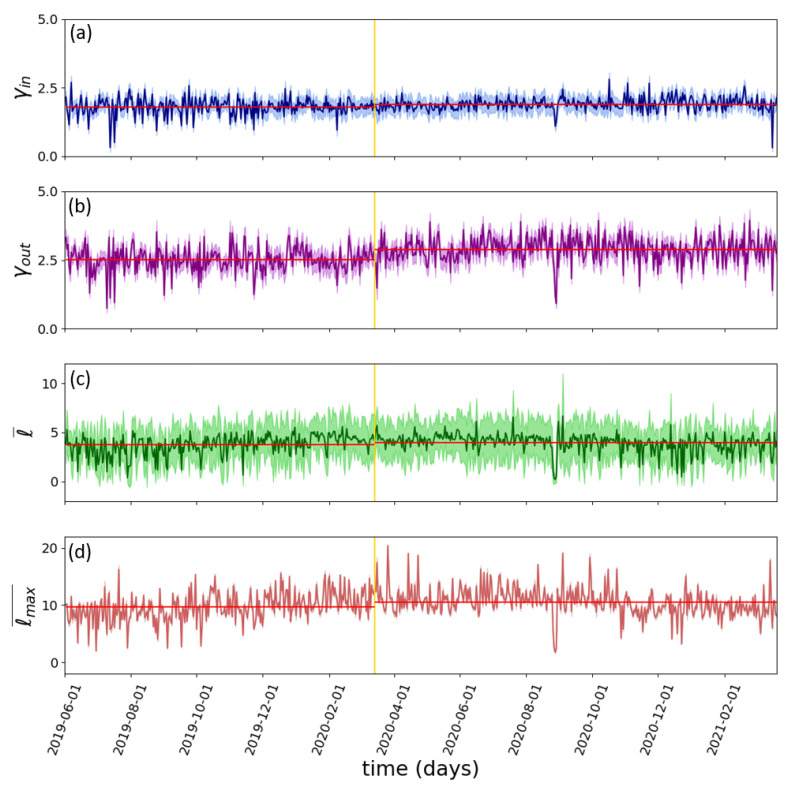
Time evolution of the scaling exponents $\gamma_{in}$ (**a**), $\gamma_{out}$ (**b**), the average shortest path length ℓ (**c**) and (**d**) the network diameter $\bar{\ell_{\max}}$. The yellow line denotes the start of the mandatory confinement ordered by the government of Spain due to the COVID-19 pandemic. Horizontal lines show the average value of each parameter before and after 15 March 2020.

**Figure 4 entropy-25-00507-f004:**
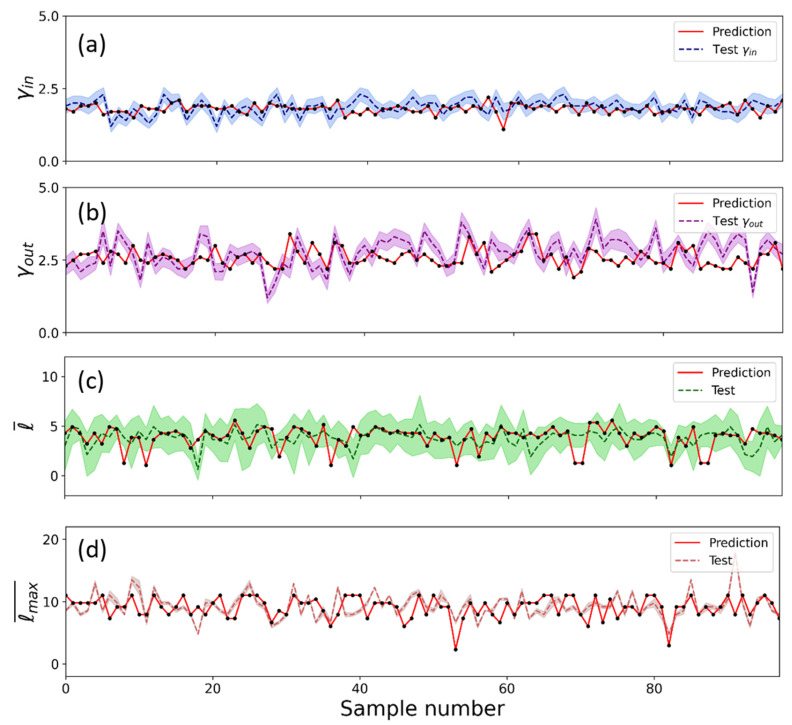
Experimental parameters (dashed colored lines) compared to those predicted by the neural network (solid red lines), for the same metrics presented in Figure 3. (**a**) $\gamma_{in}$, (**b**) $\gamma_{out}$, (**c**) the average shortest path length ℓ, and (**d**) the network diameter $\bar{\ell_{max}}$.

**Figure 5 entropy-25-00507-f005:**
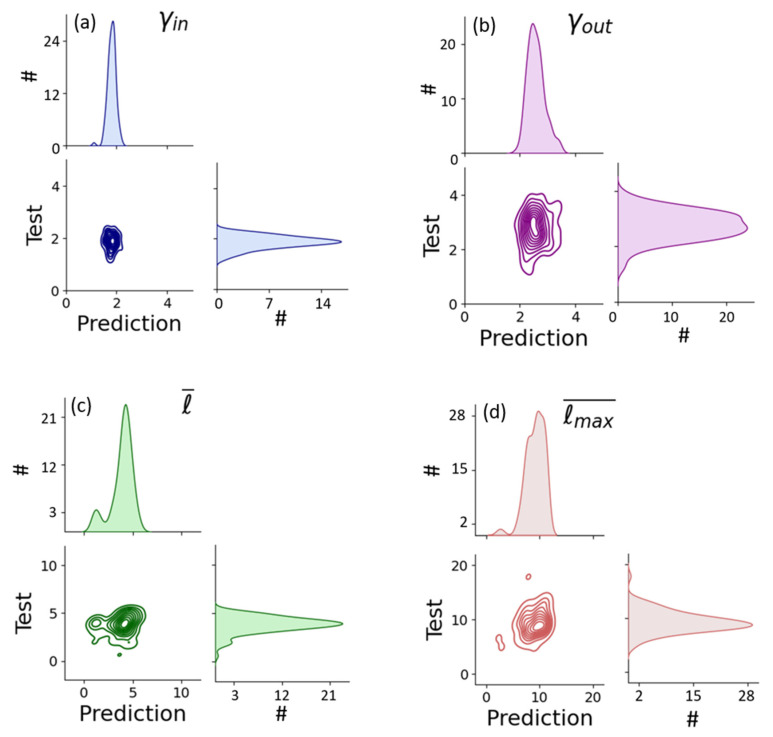
Histograms comparing the different properties ((**a**) $\gamma_{in}$, (**b**) $\gamma_{out}$, (**c**) $\bar{\ell }$, and (**d**) $\bar{\ell_{\max}}$ ), calculated using the test subset of images and their predictions using the CNN. Graphs in the diagonal represent histograms of the values found in the predict and test datasets for each parameter. Antidiagonal graphs show how similar both datasets are for each parameter. The ideal outcome would be a distribution centered in the line of the type f(x) = x.

## Data Availability

The data presented in this study are available on request from the corresponding author.

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
