# Peer review of "Artificial Intelligence Techniques Used to Extract Relevant Information from Complex Social Networks"

_entropy, 2023, doi:10.3390/e25030507_

Round 1
Reviewer 1 Report
Please see attachment.

Author Response
Referee 1: This manuscript presents an application of a Machine Learning technique to extract information from complex social networks. The subject is of interest and timely, however it lacks of precision in several aspects, and sometimes it is not clear what they are really dealing with. In particular, the manuscript could be improved by taken into account the following comments. 1) The abstract is too general. Authors should give specific information about what they are studying in this manuscript. For example, they should explicitly mention which is the novel approach (or which are its main characteristics) they have proposed to unveil the structural properties of the experimental data from commonly used networks. Which kind of data, which social networks, …. The abstract should clearly state the specific contribution of the research carried out.
Answer: The abstract has been rewritten as well as parts of the discussion/conclusions and we believe the goals of the manuscript are more clearly expressed.
Referee 1: 2) In this study authors download data from Twitter for five topics corresponding to top hashtags or keywords. However, they don´t specify which are the topics considered or the criteria to select them. How relevant are they? In twitter there are several kinds of interaction that can be considered (follow, retweet, mention,…). Which one are they considering in order to build the networks? Would the results be the same if others interactions would be considered? How robust are the results?
Answer: The criteria to select the topics was based in picking the most popular ones, in descending order. We added a section in the SI with some examples of the hashtags taken for several days. We believe this may help the reader to understand the process. The type of interactions considered were retweet, mentions and quotes, as it is specified in the first paragraph of section 2.1. In the same paragraph, it is briefly commented why we did not consider another type of interaction (the follower/friends network). We have also rewritten this part to make it clearer.
Referee 1: 3) The AI technique considered is CNN, have authors used others’ ?
Answer: Since the identification of complex network features is done through an image, it seemed logical to use a CNN by its very nature. In any case, preliminary tests were made with other types of networks such as DNN (Dense Neural Networks) and RNN (Recurrent Neural Networks), specifically LSTM and GRU but, as expected, acceptable results were not achieved with these neural types of networks. Comments on this sense have been included in section 2.3 to make it clearer.
Referee 1: 4) The term “scale-free exponent” is used in the text. Although it is clear the meaning, I am not sure if it is generally used. Authors mean that the degree distribution follow a power law with a scale free behavior. However, I consider that it could be more appropriate to refer it as “the scaling exponent”.
Answer: We agree with the reviewer that the term used could be very technical for the general reader. Following his/her suggestions we have renamed them as “the scaling exponents” when appropriate.
Referee 1: 5) Another concern is about the fact of using the values of the in and out scaling exponents corresponding to the fits to a power law of the corresponding degree distributions of the networks considered. A value of this scaling exponent could represent different networks. Another thing to clarify is if the prediction model consider simultaneously the three properties of the network considered, or just one by one.
Answer: We are aware that different network structures can have the same scaling exponents. We believe that our machine learning approach is able to recover this feature, as it has obtained for different networks similar values that match with the computed scaling exponent. This can be grasped from inspection of Figure 3. In fact, we do not try to fully describe a network by the set of this 4 parameters
Referee 1: 6) Results shown in Figure 4 need to be further explained and discussed. The figure captions is not informative enough and it is not clear what is shown in each plot of the different panels of the figures
Answer: The comments on this figure have been expanded as well as all other figures in the text trying to provide the reader with a better understanding of the issue.
Reviewer 2 Report
In this article, the authors build a network through Twitter data and continuously add labels. The authors analyze the average shortest path length, the number of nodes, in-degree and out-degree of the network in this evolution process. In addition, the authors establish a CNN model to predict the in and out scale-free exponents and average shortest path during the evolution process. The study reveals the potential of deep learning for image analysis of complex networks. I have the following comments for the authors to consider.
(1) When referring to supplementary material in the paper, it should be clearly stated in which section of the supplementary material.
(2) The structure of Figure S2 is very large, but the characters are too small. The authors should redesign it.
(3) The paper mentions Figure 2a and Figure 2b, but a and b are not marked in the illustration. Similar problems exist in other illustrations.
(4) In the last paragraph of 3.1, the author mentions the horizontal red line. However, I don't see the horizontal red line in Figure 3, but I see horizontal red lines in Figure 2. I think this paragraph actually discusses Figure 2. Are you confusing Figure 2 with Figure 3 here?
(5) Please give the analysis in the last paragraph of 3.1, how to filter and analyze the data.
(6) In 3.2, the authors should give a detailed introduction to the training process and training parameters of CNN, such as learning rate, epoch, etc.
Author Response
Referee 2: In this article, the authors build a network through Twitter data and continuously add labels. The authors analyze the average shortest path length, the number of nodes, in-degree and out-degree of the network in this evolution process. In addition, the authors establish a CNN model to predict the in and out scale-free exponents and average shortest path during the evolution process. The study reveals the potential of deep learning for image analysis of complex networks. I have the following comments for the authors to consider.
(1) When referring to supplementary material in the paper, it should be clearly stated in which section of the supplementary material.
Answer: We apologize for this. Direct references to the Figures of the supplementary materials have now been added to the main text.
Referee 2: (2) The structure of Figure S2 is very large, but the characters are too small. The authors should redesign it.
Answer: The figure has been replaced by another one that we believe better explains the architectures used for the CNN.
Referee 2: (3) The paper mentions Figure 2a and Figure 2b, but a and b are not marked in the illustration. Similar problems exist in other illustrations.
Answer: Thanks for noticing, we already corrected all figures.
Referee 2: (4) In the last paragraph of 3.1, the author mentions the horizontal red line. However, I don't see the horizontal red line in Figure 3, but I see horizontal red lines in Figure 2. I think this paragraph actually discusses Figure 2. Are you confusing Figure 2 with Figure 3 here?
Answer: Indeed, this was a misprint. We lament this mistake and corrected it in this new version.
Referee 2: (5) Please give the analysis in the last paragraph of 3.1, how to filter and analyze the data.
Answer: The data presented in Figure 3 was unfiltered. Nevertheless we adde a paragraph at the end with details about the calculation of the averages and dispersions.
Referee 2: (6) In 3.2, the authors should give a detailed introduction to the training process and training parameters of CNN, such as learning rate, epoch, etc.
Answer: Section 3.2 has been modified to include this information.
Reviewer 3 Report
An article on a very interesting topic. A solid refinement is required for the abstract.
Abstract written in a very general, unreliable way. I recommend specifying the exact purpose of the work, what methodology was used, and what results were achieved.
Introduction
In line 35, the authors write that: "In the past decade, many studies have been devoted to analyzing data from social media, especially from the website Twitter..." and what about LinkeLn, Interest in this portal is also growing. There are also studies on this topic. I definitely recommend mentioning this portal at this point.
https://scholar.google.com/scholar?hl=en&as_sdt=0%2C5&q=social+media+linkedin&btnG=&oq=social+media+linkel
I recommend that the authors provide more substantive information about what is novel about the work and what gaps are filled in the manuscript in the introduction. The sentence in line 57: "The aim is to develop a tool that could be able to compute structural graph properties in a quick and efficient way overcoming the previous drawbacks" is not clear.
The strong point of the book is the interesting methodology, the weak point is the lack of a solid review of the literature, made more so by the fact that there are a lot of publications on the subject. I recommend creating a Literature review section and fine-tuning this point. The authors additionally refer to very old studies from 1999 or 1959. I recommend removing them and focusing on the latest studies from 2022-2023. The lack of a solid analysis of the literature reduces the value of the work. The conclusion is also too broad and brief.This section also needs some refinement.
Punctuation marks should also be refined in the work.
Author Response
Referee 3: An article on a very interesting topic. A solid refinement is required for the abstract.
Abstract written in a very general, unreliable way. I recommend specifying the exact purpose of the work, what methodology was used, and what results were achieved.
Answer: The abstract, following your comments, has been rewritten as well as important parts of the discussion/conclusions to clearly state the goals of the manuscript and our results.
Referee 3: Introduction
In line 35, the authors write that: "In the past decade, many studies have been devoted to analyzing data from social media, especially from the website Twitter..." and what about LinkeLn, Interest in this portal is also growing. There are also studies on this topic. I definitely recommend mentioning this portal at this point.
https://scholar.google.com/scholar?hl=en&as_sdt=0%2C5&q=social+media+linkedin&btnG=&oq=social+media+linkel
Answer: Further literature was reviewed and articles using data from Facebook and Linkedin that could be of relevance to the text were also incorporated and commented in the introduction.
Referee 3: I recommend that the authors provide more substantive information about what is novel about the work and what gaps are filled in the manuscript in the introduction. The sentence in line 57: "The aim is to develop a tool that could be able to compute structural graph properties in a quick and efficient way overcoming the previous drawbacks" is not clear.
Answer: Following the reviewer’s suggestion, the mentioned paragraph has been rewritten and hope that now the introduction gives a clearer perspective on the work’s contribution. Significant parts of the discussion have also been rewritten following the referee’s suggestion.
Referee 3: The strong point of the book is the interesting methodology, the weak point is the lack of a solid review of the literature, made more so by the fact that there are a lot of publications on the subject. I recommend creating a Literature review section and fine-tuning this point. The authors additionally refer to very old studies from 1999 or 1959. I recommend removing them and focusing on the latest studies from 2022-2023. The lack of a solid analysis of the literature reduces the value of the work.
Answer: The mentioned studies from 1999 and 1959 by the reviewer, refer to the original papers where the algorithms mentioned in the manuscript originally appeared. These algorithms are still present and being used today, thus the inclusion of the references in the manuscript. Nevertheless, some additional references are included in the manuscript.
Referee 3: The conclusion is also too broad and brief. This section also needs some refinement.
Answer: Conclusion has been rewritten and expanded to better point out our main results and contributions.
Referee 3: Punctuation marks should also be refined in the work.
Answer: We apologize for this; text has been carefully checked in this new version.
Round 2
Author Response
Reviewer 1:
REVIEW VERSION 2
Although authors have taken into account most of the reviewer´s suggestions, still there are some minor points that they could consider to improve the manuscript.
Referee 1: 1) The results shown in Figure 1d show how the scaling exponent for the OUT degree of the network converges to a certain value as they expand the network increasing the number of hashtags considered, finding the convergence for 5 hashtags. I wander if the convergence is the same when considering the scaling exponent of the IN degree distribution. For the sake of consistence, I would suggest to provide the corresponding figure. It should be notice that the in degree concern more to the collective behavior, while the out degree to the individual actions, and accordingly their nature are different.
Answer: We included the suggested graph in Figure 1 and a brief discussion.
Referee 1: 2) In Figures 1 (b and c) it can be noticed dots with light blue color corresponding to the broad tail part of the degree distributions. It should be clarified if the color has any meaning, and if not, they should look the same.
Answer: lighter points mark those points that were not used for the linear fits. This is a common procedure as those points are endowed with a larger uncertainty. A comment has been added in section 2.2
Referee 1: 3) When discussing Figure 5, it is mentioned: “in all three cases”, while it should say four.
Answer: Corrected, thanks
Referee 1: The Seaborn plot is useful when considering more than two variables. However in the present case the labels of the axis in each panel are a little confuse, and I doubt they are correct. For example, the plots in the diagonal are Histograms, thus the vertical axis should be the number of occurrences or the frequency or probability of each value of the parameter considered for the prediction / test. Thus, the label Prediction is confused to me and the scale or digits of this axis, may be different than the prediction values than would correspond to the antidiagonal upper plot.
In summary, I think that the labels in the bottom plots of each panel are correct. However, the vertical axis of the histograms should be modified.
Concerning the antidiagonal plots, where the test and predicted values are plotted, it seems that the information provided in Prediction vs Test is the same as the test vs prediction. Accordingly, I suggest to remove one of them, and consider the organization of each panel as shown in the scheme below.
Answer: We changed this figure following the referee advise.
Referee 1: 4) In the Discussion section, it is mentioned: “As the hashtags are chosen by popularity, this presents an example of network that we consider representative of the total network. The fact that the exponent of the network converges to a given value as more subnetworks are added (Figure 1d), just confirms this hypothesis.”
This fact is only shown for the out scaling exponent. Authors should discuss if this effect is the same for the in exponent.
Answer: This was also discussed in the first comment from this referee. We are now presenting also the equivalent of Fig. 1d for the other exponent. Also, a brief description is included in the text.
Reviewer 2 Report
1. In the second paragraph of 3.2, I suggest the authors provide further explanation on the sentence 'Validation split of the train data has been set as 0.9'.
2. The tenses in the article are not uniform, some are past tense and some are present tense.
3. Supplementary material S6 is not mentioned in the main text, and I recommend that the authors confirm whether this section is required. Also, the description of S6 is incomplete, I wonder why only labels in black are considered in the final structure.
4. During the training process of CNN, some machine learning evaluation indicators should be considered to evaluate the robustness of the model.
5. Supplementary material S3 designs two CNN structures to predict parameters. Please introduce the reasons for this distinction or the analysis process for designing these two CNN structures.
Author Response
Referee 2: 1. In the second paragraph of 3.2, I suggest the authors provide further explanation on the sentence 'Validation split of the train data has been set as 0.9'.
Answer: We actually had a mistake in this paragraph. We meant that 10 % of our train data has been used for validating the results when fitting the models, that means the validation split has been set to 0.1. More details have been provided and the mistake has been fixed, thank you.
Referee 2: 2. The tenses in the article are not uniform, some are past tense and some are present tense.
Answer: We checked the text looking for these mistakes.
Referee 2: 3. Supplementary material S6 is not mentioned in the main text, and I recommend that the authors confirm whether this section is required. Also, the description of S6 is incomplete, I wonder why only labels in black are considered in the final structure.
Answer: We made sure that all sections of the SI are now properly cited in the text.
Referee 2: 4. During the training process of CNN, some machine learning evaluation indicators should be considered to evaluate the robustness of the model.
Answer: Cost functions are now included in the SI. As seen from the evolution of the cost functions during successive training epochs in the new Figure S6 in section S7 of the Supplementary Materials, the model is able to acquire the ability to estimate the values of the parameters of complex networks through their images and obtain a value close to the one calculated analytically in a much faster way. Again, our objective in this work is to show that this type of neural network is a tool that has this capability, even with a very basic architecture, with hardly any optimization of its hyperparameters. Of course (but this is beyond the scope of this paper), with optimized and more conscientious training, values could be predicted much more accurately.
Referee 2: 5. Supplementary material S3 designs two CNN structures to predict parameters. Please introduce the reasons for this distinction or the analysis process for designing these two CNN structures.
Answer: We modified the Supplementary Material to include the following discussion:
The performance of any model is highly dependent on the type of data it is trained on. Both models were tried on all four parameters ( and ) and the architecture with the best fit to the data was chosen as the most appropriate model for predicting it.
Since the objective was to prove the possibility of estimating parameters with only images of the networks keeping the model as simple as possible, only slight adjustments in the number of convolutional and pooling layers were considered for improving the observed results.
Reviewer 3 Report
Thank you for improving the manuscript.
Author Response
Referee 3: Thank you for improving the manuscript.
Answer: Thanks for your comments.